# Therapeutic Stimulation of Glycolytic ATP Production for Treating ROS-Mediated Cellular Senescence

**DOI:** 10.3390/metabo12121160

**Published:** 2022-11-23

**Authors:** Victor I. Seledtsov, Alexei A. von Delwig

**Affiliations:** Innovita Research Company, 14166 Vilnius, Lithuania

**Keywords:** reactive oxygen species, cellular senescence, glycolysis, intermittent hypoxia, blockade of respiratory chain

## Abstract

Cellular senescence is conditioned through two interrelated processes, i.e., a reduction in adenosine triphosphate (ATP) and the enhancement of reactive oxygen species (ROS) production levels in mitochondria. ATP shortages primarily influence the energy-intensive synthesis of large biomolecules, such as deoxyribonucleic acid (DNA). In addition, as compared to small biomolecules, large biomolecules are more prone to ROS-mediated damaging effects. Based on the available evidence, we suggest that the stimulation of anaerobic glycolytic ROS-independent ATP production could restrain cellular senescence. Consistent with this notion, non-drug related intermittent hypoxia (IH)-based therapy could be effectively applied in sports medicine, as well as for supporting the physical activity of elderly patients and prophylactics of various age-related disorders. Moreover, drug therapy aiming to achieve the partial blockade of respiratory chain and downstream compensatory glycolysis enhancement could prove to be useful for treating cardiovascular, neurological and hormonal diseases. We maintain that non-drug/drug-related therapeutic interventions applied in combination over the entire lifespan could significantly rejuvenate and prolong a high quality of life for individuals.

## 1. Introduction

A number of intricate mechanisms underlie cellular senescence, such as mitochondrial dysfunction, disturbances of protein epigenetic changes, telomere attrition, etc. Characteristically, age-related diseases encompass some, if not all, of those mechanisms, which give rise to cellular senescence observed in disorders such as neurodegenerative diseases, cancer, and diabetes [1]. Clearly, specific therapeutic countermeasures should be developed and explored to restrain cellular senescence and to treat age-related diseases, with many studies emphasizing the significance of mitochondrial impairments in the onset of age-related diseases. In particular, mitochondrial malfunction engenders metabolic disorders associated with an increased production of reactive oxygen species (ROS) [2]. ROS occur either as so-called free radicals with unpaired electrons, such as superoxide (O_2_^•−^) and hydroxyl radical (•OH) or as “non-radical derivatives”, such as hydrogen peroxide (H_2_O_2_). Since ROS abstract electrons from other molecules to gain stability, free radicals are constantly evolving and interacting with each other [3]. Importantly, increased ROS production has been described (i) to cause profound damages to the structure of DNA and other biomolecules, including proteins involved in oxidative phosphorylation (OXPHOS) [2], (ii) to inhibit cell cycle progression, (iii) to stimulate cellular apoptotic processes, (iv) to activate pro-inflammatory signalling pathways, (v), and (vi) to accelerate ageing processes [1]. 

In this paper, we discuss the idea that a combination of non-pharmacological and pharmacological stimulation of glycolytic adenosine triphosphate (ATP) production can be considered as a powerful strategic therapeutic approach to ensure the effective treatment of age-related diseases and prolongation of full-quality life.

## 2. Disbalance of ATP/ROS Mitochondrial Production Drives Cellular Senescence

Somatic cells are endowed with a limited number of mitotic divisions after which they lose their proliferative capacity and undergo cellular senescence characterized by pronounced phenotypic and functional alterations. Although this process affects most organelles, the most prominent changes occur in mitochondria [2]. Under normal conditions, most of the energy required for cellular functions is generated in mitochondria through ATP production by OXPHOS, which involves mitochondrial electron transport chain (ETC) complexes I-IV and ATP synthase (complex V). Electron transfer from the mitochondrial matrix to the inner membrane space generates an electrochemical gradient, which is eventually utilized by ATP synthase to produce ATP, CO_2_ and water [1]. OXPHOS is the most efficient metabolic pathway, producing approximately 36 molecules of ATP per one molecule of glucose compared to 2 ATP molecules produced during glycolysis, a mitochondria-independent cytoplasmic process that also uses glucose [4]. Under steady-state conditions, the rate of electron transport remains in equilibrium with proton translocation resulting in sufficient energy production and the minimal generation of ROS [5].

During OXPHOS, H_2_O_2_ and O_2_^•−^ are produced as byproducts in mitochondria primarily by complexes I and III of mitochondrial ETC followed by sequestration by antioxidant enzymes, including superoxide dismutase, glutathione peroxidase, glutaredoxins, thioredoxins and catalases [6]. In physiological circumstances, ROS function as signalling molecules mediating many biological responses, including cell proliferation, migration, differentiation, and gene expression. In addition, ROS serve as signalling molecules involved in autophagy, including mitophagy, which plays a pivotal role in maintaining mitochondrial quantity and quality by removing damaged or defective mitochondria [7]. 

Under pathological conditions, however, mitochondrial dysfunction has been described to cause disbalanced ATP/ROS production manifesting itself in reduced ATP and a concomitant increase in ROS production, which reflects inefficient electron transport in ETC [1]. The current paradigm suggests that the proportion of dysfunctional mitochondria increases gradually and irreversibly in the process of cellular senescence, and is accompanied by decreased mitochondrial ATP production and increased ROS levels [2]. Furthermore, disbalanced ATP/ROS mitochondrial production hinders the energy-intensive synthesis of large biomolecules, such as DNA. Mitochondrial DNA (mtDNA) was shown to be especially vulnerable to increased ROS levels due to ROS-induced damage of mtDNA repair mechanisms. In addition, ROS affects the histone methylation status at the histone methyltransferase level, thus effectively perturbating histone-dependent nuclear DNA packaging. The metabolism of ROS and Ca^2+^ is inseparably interconnected with numerous crucial control mechanisms also feeding into the development and progression of age-related diseases. Indeed, high Ca^2+^ levels within mitochondria have been described to efficiently stimulate ROS production and trigger cell death [3].

Figure 1 summarizes the proposed concept that young poorly differentiated cells are characterized by relatively high ATP and low ROS production levels accompanied by a high growth potential and low apoptotic activity of those cells. Herein, cell proliferation provides mitochondria with the critically important milieux necessary and sufficient for cell multiplication and renewal. However, under conditions of reduced ATP production paralleled with increased ROS levels, the growth and adaptive potential of mature cells are diminished, thus resulting in the up-regulation of apoptotic activity. Ultimately, old senescent cells display a critical proportion of dysfunctional mitochondria, such that those cells exhaust their proliferative and adaptive capabilities and fail to withstand apoptotic processes. 

Cellular senescence is characterized by profound alterations in the energy metabolism with an ever decreasing reliance on OXPHOS counter-balanced by increasing dependence on glycolysis. Indeed, glycolysis has been shown to be up-regulated in senescent cells in order to generate additional ATP to compensate for the inefficient energy production in dysfunctional mitochondria [2]. 

In contrast to normal cells, rapidly growing tumor cells have been shown to utilize predominantly glycolysis for ATP generation independently of oxygen availability, which is known as the Warburg effect. Generally speaking, the glycolytic pathway of ATP production is not associated with the generation of cytodestructive ROS [8]. Therefore, we maintain that decreased ROS production constitutes a pivotal factor of high growth activity and long-term survival of tumor cells. We hypothesize that in the process of evolution eukaryotic cells could have entered into a symbiotic relationship with aerobic bacteria, thus acquiring more efficient energy generating machinery. This acquisition enhanced the adaptive possibilities of cells in the context of the entire multicellular organism at the expense of cell immortality. At the same time, glycolysis continues playing a pivotal adaptive energetic role in the organism, with hypoxia and increased physical activity stimulating the glycolytic pathway of ATP production. We also suggest that glycolysis could be used to compensate for the temporary energy deficit observed in dividing cells stemming potentially from peculiar mechanisms of mitochondrial replication, segregation, distribution and inheritance during cell division. The importance of glycolysis in ATP production should theoretically increase in immune cells attracted to/present in inflammatory loci characterized by high acidity and relative oxygen deficiency due to impaired tissue microcirculation. We maintain that any physiological process occurring in organisms that require a certain level of energetic cellular autonomy should be highly dependent on the glycolytic pathway of ATP generation.

From this standpoint, we suggest that the increased glycolysis rates and waning of OXPHOS observed in senescent cells are not pathological, but rather reflect adaptive processes aiming to diminish cytodestructive ROS-mediated activity. Should it prove to be the case, putative therapy of age-related diseases should aim to primarily support and enhance the cytoplasmic glycolytic pathway of ATP production, which theoretically would alleviate the functional burden on mitochondria and restrain their senescence in the long run.

## 3. Non-Drug-Related Treatments to Stimulate Glycolysis

Hypoxia represents a natural evolutionary proven inducer of glycolytic activity in eukaryotic cells that lead to multifaceted effects on the cellular metabolism. Oxygen deficiency is known: (i) to activate glycolytic enzymes, (ii) to activate hypoxia-inducible factor-1 (HIF-1) and nuclear factor erythroid 2–related factor 2 (Nrf2), and (iii) to enhance erythropoietin (EPO) (a powerful naturally occurring intermittent hypoxia-induced neuroprotectant) and pro-angiogenic vascular factor production [9].

Intermittent hypoxia (IH) traditionally includes periodic and alternating cycles of hypoxia and normoxia. With the development and widespread availability of devices inducing systemic or local hypoxic environments (e.g., hypobaric chambers, hypoxia rooms and tents, hypoxicators, or pneumatic cuffs), IH has gained considerable popularity as an effective training modality for a variety of professional athletes, as well as a non-pharmacological approach to prevent and treat various diseases [10]. Importantly, hypoxia-mediated effects on the body overlap with the outcomes of nonpharmacological interventions, such as breathing yoga, exercise and caloric restriction, which are known to reduce oxidative damage and inflammation and improve health, lifespan, and cognitive functions [6]. Not surprisingly, IH has been used extensively for treating a variety of clinical disorders, including chronic lung diseases, bronchial asthma, hypertension, diabetes mellitus, Parkinson’s disease, emotional disorders, and radiation toxicity, as well as for the prophylaxis of certain occupational diseases and in sports medicine [11].

IH was shown to up-regulate the activity of antioxidant enzymes, which make cells less sensitive to the cytodestructive ROS-mediated activity. Importantly, the IH-induced expression of glycolytic and antioxidant genes is characterized by a certain momentum, such that it continues for a while after the cessation of a hypoxic episode [12]. Therefore, IH is likely to enhance cellular protection from subsequent oxidative stress. In contrast to IH, constant long-term hypoxia down-regulates antioxidant-mediated cell protective mechanisms augmenting cell sensitivity to oxidative stress-associated toxicity [13]. In general, the published data suggest that IH is capable of restraining the age-related accretion of ROS-mediated pathological processes, thus offering itself as a promising non-pharmacological intervention strategy for preventing and treating various age-related diseases.

## 4. Drug Related Treatments to Stimulate Glycolysis

Multiple studies conducted to date have indicated that mild mitochondrial stress associated with the inhibition of OXPHOS complexes (and, consequently, with compensatory glycolysis enhancement) could induce adaptive stress responses, thus promoting health and delaying the development of age-related degenerative diseases. In particular, mutational studies designed to inhibit mitochondrial respiratory chain activity documented 20–300% increases in the mean adult lifespan in *C. elegans* [6,14]. Similar effects on longevity modulation have been achieved through the RNA interference (RNAi)-mediated reduction in ETC component machinery expression. However, the complete ablation of major ETC subunits resulted in severe phenotypes and shorter lifespan, indicating that only a mild decrease in ETC activity was beneficial [14].

During in vivo experiments, decreased ROS production and increased longevity (by 30%) were observed in a murine model characterized by the decreased expression of proteins involved in ETC machinery, with particular reference to matrix arm subunits of complex I [15,16]. Additionally, partial inhibition of complex IV and cytochrome c oxidase activity was shown to increase longevity in mice and conferred protection from neurodegeneration [17]. Severe deficiency in complex IV or mild deficiency in complex III expression in neurons provided neuroprotective effects in an APP/PS1 murine model of Alzheimer’s disease [18], while the inhibition of complex V was shown to increase neuronal survival in response to toxic agents in vitro [19]. Furthermore, the clinical application of molecules causing mitochondria depolarization with a downstream ATP decrease and OXPHOS uncoupling has been considered as a promising therapeutic strategy for combatting aging, obesity, neurodegeneration, and cancer. However, OXPHOS uncouplers displayed multiple off-target effects and toxicity with potential limitations to their clinical usage [6].

In general, the adaptive response to energetic stress via activation of AMP-activated protein kinase (AMPK) constitutes an important molecular mechanism linked to life-extending interventions associated with the mild inhibition of OXPHOS across species [20]. Furthermore, AMPK activation was shown to enhance autophagy, thus mediating the removal of damaged organelles and misfolded proteins to improve cellular protein homeostasis (proteostasis) [21], while increasing the production of “young” mitochondria via biogenesis.

At present, there are medicinal drugs available that can reduce energetic and ROS-producing mitochondrial activity, such as Metformin which is widely used for treating type 2 diabetes. Indeed, Metformin was demonstrated to inhibit mitochondrial complex I of the respiratory chain and activate AMPK among other multiple targets, such as NF-κB signalling [22]. Being positively charged, Metformin accumulates in mitochondria reaching up to 1000-fold higher concentrations than that in the extracellular medium, which is sufficient for complex I inhibition [23]. Another promising therapeutic compound in this respect is Resveratrol, which has been shown to inhibit the activity of mitochondrial complexes I, III, and V [6,24]. Similar to Metformin, Resveratrol stimulates key signalling pathways, including antioxidant defences, and reduction in inflammation via NF-kB signalling inhibition, thus preventing the deleterious effects triggered by oxidative stress [25]. Progesterone is a neurosteroid hormone that blocks mitochondrial complex I and displays protective effects for the function and vitality of neural cells. In addition, Progesterone was shown to reduce inflammation, oxidative stress, and apoptosis, while promoting DNA repair [26]; it was thus revealed to be a promising candidate for treating brain injury. Recent studies have identified other ETC inhibitors possessing a wide range of biological properties, including antioxidant, anticancer, anti-inflammatory, and cardio- and neuro-protective effects [6,27]. Moreover, many polyphenols, vitamins, and trace elements are known to have antioxidant properties [28]; however, their biological activity is dependent on continuous drug intake. It must be stressed that high doses of these compounds could affect/inhibit both deleterious and positive ROS-mediated activity.

## 5. Conclusions

Currently, the mitochondrial theory of aging is the most reasonable and generally accepted. In this respect, most scientific efforts have focused on inducing constantly senescent mitochondria to effectively produce ATP [2]. However, in this paper we speculate that any artificial promotion of functional mitochondrial activity is fraught with an undesired up-regulation of cytodestructive ROS production, which in fact would facilitate cellular senescence. We propose a more tenable physiological approach for restraining and retarding cellular senescence based on the stimulation of the glycolytic pathway of ATP production. This strategy would relieve the functional burden and stress from mitochondria as far as possible, thus effectively reducing mitochondrial production of cytodestructive ROS. The development and clinical application of complex non-drug/drug-related therapeutic interventions aiming to promote glycolytic ATP production will allow for significant progress in both prevention and treatment of various age-related diseases.

## Figures and Tables

**Figure 1 metabolites-12-01160-f001:**
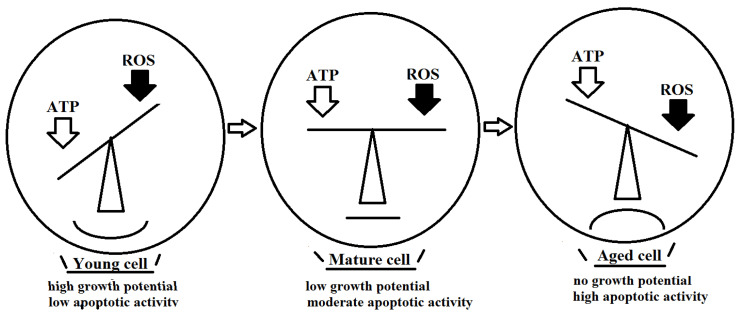
Relative interrelationships between ATP and ROS production in the course of cellular senescence. See description in the text.

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
