# Peer review of "Therapeutic Stimulation of Glycolytic ATP Production for Treating ROS-Mediated Cellular Senescence"

_metabolites, 2022, doi:10.3390/metabo12121160_

Round 1
Reviewer 1 Report
This viewpoint article is addressed to the interconnection between cellular metabolism and senescence. Authors discuss the role of mitochondria healthness in the production of ROS that damage intercellular components and promote the senescence. The glycolisys produces ATP not so effectively like mitochondria but there is no ROS accumulation that helps cells don't senecsence. Authors hypothesise that it may be applicable to the organism and stimulation of the glycolysis together with the inhibition of mitochondria ATP production may be used as a longevity strategy as well as a prevention and treatment of age-related diseases. It is very interesting viewpoint that is argumented very good with appropriate references and examples that are used already for diseases treatment. However, I think that authors should discuss more detailed the physiological cases of the switching cellular metabolism from OXPHOS to glycolisys that occur when cells increase the proliferation rate in cancer transformation, during immune response, regeneration and other. This manuscript doesn't discuss the negative consequences that proposed therapy/treatmant may have. This minor changes should improve the manuscript.
Author Response
Thank you very much for your invaluable comments and suggestions that allowed us to greatly improve our viewpoint paper.
In full compliance with the reviewer’s suggestion, we emphasized the role of glycolysis in adaptive physiological processes. In particular, we stated that (line 113-123):
At the same time, glycolysis continues playing a pivotal adaptive energetic role in the organism, with hypoxia and increased physical activity stimulating glycolytic pathway of ATP production. We also suggest that glycolysis could be used to compensate for temporary energy deficit observed in dividing cells stemming potentially from peculiar mechanisms of mitochondrial replication, segregation, distribution and inheritance during cell division. The importance of glycolysis in ATP production should theoretically increase in immune cells attracted to/present in inflammatory loci characterized by high acidity and relative oxygen deficiency due to impaired tissue microcirculation. We maintain that any physiological process occurring in the organisms that requires certain level of energetic cellular autonomy should be highly dependent on glycolytic pathway of ATP generation.
We also mentioned negative consequences that proposed therapy/treatmant may have (lines 181-183):
However, OXPHOS uncouplers displayed multiple off-target effects and toxicity with potential limitations to their clinical usage [6].
In addition, we added a new statement that emphasizes limitations to the usage of such compounds as polyphenols, vitamins, and trace elements (lines 208-209):
It must be stressed that high doses of these compounds could affect/inhibit both deleterious and positive ROS-mediated activity.
Reviewer 2 Report
In this manuscript, the authors introduced the data to support the idea in favor of both non-pharmacological and pharmacological stimulation of glycolytic ATP production as an effective strategic therapeutic approach to ensure effective treatment of age-related diseases and prolongation of full-quality life. I think that the topic of this manuscript is interesting. However, in my opinion, this manuscript should be summarized as review article after increasing the contents (such as the basic information about cellular senescence) and references.
Minor comments:
1.What is HT?.
2. There are many careless mistakes. The authors should check the manuscript carefully.
Author Response
We stress that the main purpose behind this viewpoint (as stated by reviewer No. 1) paper was to collect evidence and express our opinion based on the current paradigm of the role of ROS in cellular senescence with particular reference to non-pharmacological and pharmacological stimulation of glycolytic ATP production as an effective strategic therapeutic approach to ensure effective treatment of age-related diseases and prolongation of full-quality life. There is a multitude of excellent reviews on the various aspects of cellular senescence in the open literature, such that we decided not to overload this concise and up to the point opinion paper with yet another review of the subject. We think that this approach would lengthen the MS and most importantly obscure the main idea.
We also thank the reviewer for carefully reading the MS and spotting a typo (HT, which should read IH, Intermittent hypoxia). The appropriate amendments have been made.
We followed the reviewer’s advice and re-checked our MS for mistakes.
Once again, we thank the Editor and reviewers for professional and productive comments and suggestions, and we hope that the amended MS will be accepted for publication.
Round 2
Reviewer 2 Report
Thank you for responding to my comments.
I am satisfied with the authors' response.